# Quantum gravitational decoherence from fluctuating minimal length and deformation parameter at the Planck scale

Luciano Petruzziello [1,2] & Fabrizio Illuminati [1,2]

Schemes of gravitationally induced decoherence are being actively investigated as possible mechanisms for the quantum-to-classical transition. Here, we introduce a decoherence process due to quantum gravity effects. We assume a foamy quantum spacetime with a fluctuating minimal length coinciding on average with the Planck scale. Considering deformed canonical commutation relations with a fluctuating deformation parameter, we derive a Lindblad master equation that yields localization in energy space and decoherence times consistent with the currently available observational evidence. Compared to other schemes of gravitational decoherence, we find that the decoherence rate predicted by our model is extremal, being minimal in the deep quantum regime below the Planck scale and maximal in the mesoscopic regime beyond it. We discuss possible experimental tests of our model based on cavity optomechanics setups with ultracold massive molecular oscillators and we provide preliminary estimates on the values of the physical parameters needed for actual laboratory implementations.

[1] Dipartimento di Ingegneria Industriale, Università degli Studi di Salerno, Fisciano, (SA), Italy. [2] INFN, Sezione di Napoli, Gruppo collegato di Salerno, Fisciano, (SA), Italy. ✉email: lupetruzziello@unisa.it; filluminati@unisa.it

Following early pioneering studies[1–4], the investigation of the quantum-to-classical transition via the mechanism of decoherence has become a very active area of research, both experimentally and theoretically, playing an increasingly central role in the research area on the foundations of quantum mechanics (QM) and the appearance of a classical world at the macroscopic scale, as one may gather, e.g., from the many excellent existing reviews on the subject (i.e., see for instance refs. [5–13] and references therein). In broad terms, decoherence appears to be due to the inevitable interaction and the ensuing creation of entanglement between a given quantum system and the environment in which it is embedded. Indeed, no quantum system can truly be regarded as isolated, and the entanglement shared with the environment significantly affects the outcome of local measurements, even in the circumstance in which "classical" disturbances (such as dissipation and noise) can be neglected. Decoherence has been corroborated over the years by several experimental observations, among which it is worth mentioning its first detection obtained via cavity QED[14,15], as well as the various laboratory tests involving superconductive devices[16], trapped ions[17] and matter-wave interferometers[18].

Apart from the domains of condensed matter and atomic physics, decoherence has been studied in more exotic and extreme contexts, ranging from particle physics to cosmological large-scale structures. For instance, in the framework of particle physics, decoherence has been investigated at the quantum field theory level in the regime of nontrivial interactions, both at zero and finite temperature[6,19,20], and including lepton mixing and oscillations[21]. These studies and related ones have led to the opportunity of predicting new physics phenomenology, especially in connection with entanglement degradation in non-inertial frames and in the presence of strong gravitational fields[22].

In principle, as in the above examples, the main features of each decoherence mechanism depend significantly on the type of interaction with the environment. On the other hand, in an attempt to explain wave function collapse and reduction to pointer states in general terms, Ghirardi, Rimini and Weber (GRW) proposed the existence of a universal decoherence mechanism and spontaneous localization with the introduction of just two model-dependent parameters, namely the collapse rate and the localization distance[23] (for important developments in GRW-like models, we refer the reader to refs. [24–30]). Inspired by the original GRW suggestion, Diosi[31] and Penrose[32] singled out classical Einstein gravity as the possible overarching interaction responsible for the loss of coherence in GRW theory, thus refining the previous intuition and fixing the free parameters in the GRW scheme of the quantum-to-classical transition. Since the innovative Diosi–Penrose insight, much effort has been devoted to experimentally test the gravitational realization of the GRW mechanism. In this regard, the attention has been mainly focused on gravitational time dilation[33], optomechanics[34] and quantum clocks[35]. Very recently, a crucial experiment has demonstrated that the parameter-free Diosi–Penrose model based on classical Einstein gravity fails to account for the correct rates of radiation emission due to quantum state reduction, thus ruling it out as a feasible candidate for the implementation of the GRW decoherence mechanism[36].

The experimental falsification of the Diosi–Penrose version of GRW theory can be considered an important milestone as well as a new starting point. As a matter of fact, among the various ideas that revolve around the concept of gravitational decoherence (i.e., see for example refs. [37–44]; for a comprehensive review and a more detailed list of references, see e.g., ref. [45]), a significant portion deals with the high-energy regime in which quantum and gravitational effects are deemed to be comparably important. Although a complete and consistent theory of quantum gravity is still lacking, all the different current candidates including string theory, loop quantum gravity, noncommutative geometry and doubly special relativity, predict the existence of a minimal length at the Planck scale. An immediate consequence of this common aspect is the breakdown of the Heisenberg uncertainty principle (HUP), whose most famous formulation conveys that spatial resolution can be made arbitrarily small with a proper energetic probe. Therefore, as firstly derived in the framework of string theory[46] and strongly supported by thought experiments on micro black holes afterwards[47], at the Planck scale the HUP must be superseded by a generalized uncertainty principle (GUP), whose minimal one-dimensional expression reads

$$\Delta X \Delta P \geq \frac{\hbar}{2} \left( 1 + \beta \, \ell_{\mathrm{p}}^2 \, \frac{\Delta P^2}{\hbar^2} \right) , \tag{1}$$

where $\beta$ denotes the so-called deformation parameter and $\ell_{\mathrm{p}}$ is the Planck length. Starting from the above basic picture, the GUP has been discussed in the most disparate settings, from QM and quantum field theory[48–54] to noncommutative geometry and black hole physics[55–60].

In this work, we introduce a model accounting for a universal quantum-gravitational decoherence process. The model assumes deformed canonical commutation relations (DCCRs) leading to the deformed uncertainty relation (1) that accounts for the presence of a minimal length scale. Furthermore, the model assumes the minimal length scale to be a fluctuating quantity, induced by a conjectured foamy character of quantum spacetime, whose average magnitude is fixed at the Planck length scale. As a consequence, the deformation parameter $\beta$ that enters in the modified canonical momentum operator and in the DCCRs is promoted to a random quantity as well. As it is customary in the analysis of signatures of quantum gravitational effects at the classical macroscopic scale[61], in the following we treat the contribution associated to the GUP as a small perturbation. We then show how to derive a master equation of the Lindblad–Gorini–Kossakowski–Sudarshan form[62,63] for the averaged quantum density matrix out of the corresponding $\beta$-deformed stochastic Schrödinger equation for the quantum state vector, and we study the physical consequences of such open quantum state dynamics. In light of the apparent experimental confutation of the Diosi–Penrose model, our proposal may be regarded as a possible alternative universal decoherence mechanism, provided that a set of reasonable assumptions holds. At the same time, a laboratory test of our model and its predictions would also yield an indirect evidence for the quantum nature of the gravitational interaction. Along these lines, building on a universal decoherence measure introduced in ref. [34], we investigate a scheme for laboratory tests involving a cavity optomechanical system equipped with heavy molecular structures. We then provide numerical estimates and show that such a setup can probe our predictions with a high degree of accuracy.

## Results

**Deformed commutation relations, generalized uncertainty principle and modified Schrödinger dynamics.** In order to define a consistent quantum mechanical theory in Hilbert space from Eq. (1), one can start from the $\beta$-deformed canonical commutation relation, as shown in ref. [64]. In compliance with the above discussions, we can then consider the deformation of the commutator (and, in turn, the modification of the uncertainty relations) as the signature of the quantum gravitational

environment in which the system under examination is embedded. Notice that this kind of approach departs from refs. [42] and [43] which treat gravity as a classical decoherence source; at the same time, it also differs from ref. [41] that is based on perturbative quantum gravity.

For mirror-symmetric states, it is easy to prove that the uncertainty relation (1) can be straightforwardly derived from the following commutator:

$$\left[\hat{X}, \hat{P}\right] = i\hbar \left(1 + \beta\, \ell_{\mathrm{p}}^2 \frac{\hat{P}^2}{\hbar^2}\right), \qquad (2)$$

where the capital letters label the high-energy position and momentum operators, which are basically different from the usual (low-energy) ones of ordinary QM[51]. In the three-dimensional case, the most general counterpart of the previous expression that preserves rotational isotropy is given by[65]:

$$\left[\hat{X}_j, \hat{P}_k\right] = i\hbar \left(\delta_{jk} + \beta\, \ell_{\mathrm{p}}^2 \frac{\hat{P}^2}{\hbar^2}\, \delta_{jk} + \beta'\ell_{\mathrm{p}}^2 \frac{\hat{P}_j \hat{P}_k}{\hbar^2}\right), \qquad j, k = \{1, 2, 3\}, \tag{3}$$

where $\hat{P}^2 = \sum_k \hat{P}_k^2$. As frequently done in the literature[61,64], in the following we will consider the case $\beta' = 2\beta$; such a choice guarantees that the spatial geometrical structure is left commutative up to $\mathcal{O}(\beta, \beta')$. As a matter of fact, should a different selection be made for $\beta'$, we would have

$$\left[\hat{X}_i, \hat{X}_j\right] = i\hbar\, \frac{2\beta - \beta' + (2\beta + \beta')\beta\hat{P}^2}{1 + \beta\hat{P}^2} \left(\hat{P}_j \hat{X}_i - \hat{P}_i \hat{X}_j\right), \quad (4)$$

which is subtler to handle, but eventually does not affect the final outcome, since the position operator does not play a relevant rôle in the upcoming discussions. It is worth remarking that from Eq. (3) we can obtain an isotropic minimal uncertainty in position which is proportional to the Planck length, i.e., $\Delta X \simeq \sqrt{\beta}\, \ell_{\mathrm{p}}$.

As already pointed out above, in light of Eq. (3) it is not possible to simply recover the position and momentum operators of ordinary QM, since they need to be properly modified so as to account for a non-vanishing value of the deformation parameter $\beta$. A suitable choice for these operators that complies with the DCCRs requires the identification:

$$\hat{X}_j = \hat{x}_j + \mathcal{O}(\beta^2), \qquad \hat{P}_k = \left(1 + \beta\, \ell_{\mathrm{p}}^2 \frac{\hat{p}^2}{\hbar^2}\right)\hat{p}_k, \qquad (5)$$

where $\hat{x}_j$ and $\hat{p}_k$ satisfy the original HUP. The full expansion of $\hat{X}_j$ lies beyond the scope of the present paper; a more detailed treatment of these higher-order terms can be found in refs. [51]. For the sake of completeness, it must be noticed that, due to the existence of a minimal uncertainty in position, the operator $\hat{X}_j$ is no longer self-adjoint, but only symmetric[64]. Consequently, it is inappropriate to speak of position eigenstates, since the ensuing uncertainty is always non-vanishing. However, by relying on $\Delta X \neq 0$, one can build the so-called "quasi-position" space, which is made up of non-orthogonal states that represent the counterpart of the Dirac distribution $\delta(x - x')$ appearing in the standard QM framework. By thoroughly investigating the above setting, interesting phenomenological implications may be deduced even for simple problems, such as the particle in a box and the potential barrier[66].

Collecting the above results, we can now seek the proper generalization of the Schrödinger time evolution equation in order to include the effects due to the DCCRs and the GUP. Starting from (5), it is immediate to achieve the modified quantum mechanical evolution equation for pure quantum states

$|\psi\rangle$ in the form

$$i\hbar\, \partial_t |\psi\rangle = \left[\left(1 + \beta\ell_{\mathrm{p}}^2 \frac{p^2}{\hbar^2}\right)^2 \frac{p^2}{2m} + V(\mathbf{x})\right]|\psi\rangle, \qquad (6)$$

where we have omitted the hats on the operators to streamline the notation. By introducing $H_0 = p^2/2m$ and $H = H_0 + V$, the previous equation can be cast in the form

$$i\hbar\, \partial_t |\psi\rangle = \left(H + H_\beta\right)|\psi\rangle, \qquad (7)$$

where

$$H_\beta = 4\, \frac{\beta\, m\, \ell_{\mathrm{p}}^2}{\hbar^2}\, H_0^2 \left(1 + \frac{\beta\, m\, \ell_{\mathrm{p}}^2}{\hbar^2}\, H_0\right). \qquad (8)$$

The magnitude of the deformation parameter $\beta$ is commonly estimated to be of order unity[46,49,55,56,58,59]. From a more general perspective, $\beta$ may be regarded as a dynamical variable[67] whose sign in several significant works is taken as either positive[46,47,49,56] or negative[55,57,59,60]. Note that, for the case $\beta < 0$, one can come across a "classical" regime at the Planck scale, that is, $\Delta X \Delta P \geq 0$. This possibility is predicted not only by the GUP, but also by doubly special relativity[68] as well as the cellular automaton formulation of QM due to 't Hooft[69]. Furthermore, the functional analysis of the position and momentum operators can be rigorously carried out also in this context, as shown in ref. [57].

The above wide spectrum of estimates is compatible with the possibility that spacetime fluctuates as the Planck scale is approached, as predicted by major theoretical frameworks, such as the quantum foam scenario[70–72] and loop quantum gravity[73–75]. In turn, this suggests that space-time fluctuations fix the minimal length scale only on average, thereby making the associated deformation parameter itself a fluctuating quantity. Equipped with a random $\beta$, Eq. (7) is promoted to a (linear) stochastic Schrödinger equation. Since the fluctuations in $\beta$ should be related to the fluctuations of the metric tensor near the Planck threshold[71,72], within the framework of non-relativistic QM it is natural to assume $\beta$ to be a Gaussian white noise with a fixed mean and with a sharp auto-correlation so that

$$\beta = \sqrt{t_{\mathrm{p}}}\, \chi(t), \qquad \langle\chi(t)\rangle = \bar{\beta}, \qquad \langle\chi(t)\, \chi(t')\rangle = \delta(t - t'), \tag{9}$$

where $\langle \dots \rangle$ denotes an average over fluctuations whose intensity is provided by the Planck time $t_{\mathrm{p}}$ and $\bar{\beta}$ is the fixed mean. The consistency of our ansatz is supported by the observation that a white-noise process implies a large disproportion between the typical timescale of free motion and the auto-correlation time of the stochastic fluctuations. In addition, since quantum gravitational effects are expected to become maximally relevant at the Planck scale, the identification of $t_{\mathrm{p}}$ as the correct time reference is tailor-made. This is a reasonable choice to describe Planck-scale effects but it is not the only one available, as one could always set the intensity of the fluctuations at the value $\alpha\, t_{\mathrm{p}}$, where $\alpha$ would then be the remaining overall free parameter of the theory. In the following, we assume the previous parameter to be of order unity: $\alpha \simeq 1$; by this choice, we assume the intensity of the fluctuations to be roughly given by the Planck time. Bearing this in mind, in order to make our model parameter-free, we need to fix the value of the constant mean appearing in Eq. (9). The original finding in the context of string theory concerning the magnitude of $\beta$ being of order unity has since been confirmed by a large body of works starting from different frameworks; therefore, it is legitimate to set $\langle\beta\rangle = 1$. Consequently, we obtain $\bar{\beta} = 1/\sqrt{t_{\mathrm{p}}}$, by virtue of which we unambiguously identify our stochastic quantity.

Finally, from the $\beta$-deformed stochastic Schrödinger equation (7) for the state vector $|\psi\rangle$, it is straightforward to derive the ensuing $\beta$-deformed stochastic Liouville-von Neumann equation for the dynamics of the quantum density matrix $\varrho = |\psi\rangle\langle\psi|$, that is

$$\partial_t \varrho = -\frac{i}{\hbar}\left[H + H_\beta, \varrho\right] . \tag{10}$$

Equation (10) is the starting point for the analysis of the decoherence dynamics.

**Fluctuating deformation parameter and induced master equation.** Recalling the above hypothesis on the stochastic nature of the deformation parameter $\beta$ entering in the DCCRs and the associated GUP, in the following we obtain a master equation in Lindblad form for the quantum density matrix averaged over the fluctuations of $\beta$. To this aim, we begin by working in the interaction picture via the standard unitary transformation

$$|\psi\rangle = e^{-\frac{iHt}{\hbar}}|\tilde{\psi}\rangle , \tag{11}$$

which allows us to rewrite Eq. (7) as

$$i\hbar\,\partial_t|\tilde{\psi}\rangle = \tilde{H}_\beta|\tilde{\psi}\rangle , \qquad \tilde{H}_\beta = e^{\frac{iHt}{\hbar}} H_\beta\, e^{-\frac{iHt}{\hbar}} . \tag{12}$$

Consequently, after having restored the hidden dependence on the time $t$, the Liouville-von Neumann equation transforms accordingly as

$$\partial_t\tilde{\varrho}(t) = -\frac{i}{\hbar}\left[\tilde{H}_\beta(t), \tilde{\varrho}(t)\right] , \tag{13}$$

which has the formal exact solution

$$\tilde{\varrho}(t) = \tilde{\varrho}(0) - \frac{i}{\hbar}\int_0^t \left[\tilde{H}_\beta(t'), \tilde{\varrho}(t')\right] dt' . \tag{14}$$

By inserting Eq. (14) into (13), one finds

$$\partial_t\tilde{\varrho}(t) = -\frac{i}{\hbar}\left[\tilde{H}_\beta(t), \tilde{\varrho}(0)\right] - \frac{1}{\hbar^2}\int_0^t \left[\tilde{H}_\beta(t), \left[\tilde{H}_\beta(t'), \tilde{\varrho}(t)\right]\right] dt' , \tag{15}$$

where in the integral in the r.h.s. we have made use of the Born-Markov approximation to let $\varrho$ depend on $t$ and not on $t'$.

In order to provide the correct description in the mean of the quantum stochastic process under examination, we introduce the density matrix $\rho = \langle\varrho\rangle$ averaged over the fluctuations of the deformation parameter $\beta$. We remark that, being a convex combination of projectors, the average density matrix $\rho$ is in general a mixed state subject to a non-unitary time evolution and hence to dissipation and decoherence. Bearing this in mind, averaging Eq. (15) by means of Eqs. (9) and keeping only the lowest-order contributions that do not exceed $\mathcal{O}(\beta^2)$, one has

$$\partial_t\tilde{\rho}(t) = -\sigma\int_0^t \langle\chi(t)\chi(t')\rangle\left[\tilde{H}_0^2(t), \left[\tilde{H}_0^2(t'), \tilde{\rho}(t)\right]\right] dt' = -\sigma\left[\tilde{H}_0^2(t), \left[\tilde{H}_0^2(t), \tilde{\rho}(t)\right]\right] , \tag{16}$$

where

$$\sigma = \frac{16\,m^2\,\ell_{\rm p}^4\,t_{\rm p}}{\hbar^6} . \tag{17}$$

It is worth noting that the intermediate result in Eq. (16) coincides with the one that is obtained exploiting the cumulant expansion method[76], as applied in different contexts[38]. Finally, the Schrödinger representation for the average density matrix $\rho$ is recovered by applying the transformation

$$\rho(t) = e^{-\frac{iHt}{\hbar}}\,\tilde{\rho}(t)\,e^{\frac{iHt}{\hbar}} . \tag{18}$$

Assuming free motion ($V = 0$), the Lindblad-type master equation for the averaged density matrix $\rho$ reads

$$\partial_t\rho(t) = -\frac{i}{\hbar}\left[H_0, \rho(t)\right] - \sigma\left[H_0^2, \left[H_0^2, \rho(t)\right]\right] . \tag{19}$$

In the previous expression, the so-called dissipator is handily identified as the second term in the r.h.s. The Lindblad form of the master equation assures that the dynamical map is completely positive and trace preserving, thereby describing a *bona fide* quantum open system dynamics[77,78]. As required, in the limit $\sigma \approx 0$ we recover the standard Liouville-von Neumann equation and the corresponding unitary dynamics. Furthermore, looking at Eq. (19), we see that one does not have to be concerned with the effects due to spatial non-commutativity; indeed, the position operator does not appear at all in the above equation, since the GUP correction only affects the momentum, which is the sole physical quantity that enters in $H_0$.

From Eq. (19), we can also draw another interesting aspect by comparing the aforesaid expression with the outcome of previous works centered around an analogous issue. Indeed, in the dissipator the unperturbed Hamiltonian appears as $H_0^2$, whereas in related treatments on gravitational decoherence (i.e., refs. [39,79]) such dependence is linear. As a matter of fact, the present framework revolves around the existence of a minimal length $\ell_{\rm p}$ whilst in refs. [39,79] the original ansatz addresses the presence of a fundamental minimal time lapse. In principle, the two approaches might be reconciled provided that one resorts to a fully relativistic generalization of the DCCRs and of the GUP[51] to treat space and time on the same footing. However, in that case we would have to deal with two free parameters. Moreover, if we want to leave spacetime isotropy as well as the Poincaré algebra untouched, the choice of the aforementioned parameters is such that the non-relativistic limit of the new DCCRS and of the new GUP significantly differs from Eq. (3) in the values of $\beta$ and $\beta'$[80]. In addition to that, under these circumstances we would have to work with fields instead of single-particle states, and it is not straightforward or even well-defined how to proceed in such an extremely complex scenario.

From the above analysis, we can conclude that, in order for quantum gravity to affect quantum coherence, one needs two basic ingredients: the existence of a minimal length at the Planck scale and a fluctuating GUP deformation parameter. In the following, we will study some of the most relevant quantities related to quantum gravitational decoherence as described by Eq. (19), specifically the entropy production and the decoherence time. In addition, we will discuss how the latter depends on the size and the characteristic energy scales for a variety of macroscopic and microscopic systems.

**Physical results: entropy variation and decoherence time.** In this section, we study the time evolution of the linear entropy and the quantum state purity, and we estimate the decoherence time for different physical systems. In particular, we find that the quantum gravitational decoherence associated with the GUP (1) entails a strong localization in energy space.

In what follows, we consider the linear entropy, defined as[81]:

$$S(t) = 1 - \text{tr}\left(\rho^2(t)\right) , \tag{20}$$

with $\text{tr}(\rho^2(t))$ being the trace of the squared density matrix, that is the quantum state purity. Multiplying Eq. (20) by the constant factor $d/(d-1)$, with $d$ being the Hilbert space dimension, the linear entropy can be normalized in such a way that $S(t) \in [0, 1]$. Besides corresponding to the state mixedness, the linear entropy is also strictly connected to the von Neumann entropy, since the former represents the leading term of the latter in the Mercator

logarithmic series expansion[82] around the pure state condition $\rho^2 = \rho$[83]. The time evolution of the linear entropy reads

$$\partial_t S(t) = -\partial_t \text{tr}(\rho^2(t)) = -2 \text{ tr}(\rho(t)\partial_t\rho(t)) . \quad (21)$$

Inserting Eq. (19) into the above expression and omitting the explicit time dependence, we have

$$\partial_t S = \frac{2i}{\hbar} \text{ tr}(\rho[H_0, \rho]) + 2 \sigma \text{ tr}(\rho[H_0^2, [H_0^2, \rho]]) . \quad (22)$$

The first term in the r.h.s. identically vanishes due to the cyclic property of the trace, whereas the second contribution arising from the GUP must be manipulated a bit further. After some straightforward but tedious algebra, Eq. (22) can be recast in the form

$$\partial_t S = 2 \sigma \left[ 2 \text{ tr}(\rho H_0^4 \rho) - 2 \text{ tr}\left((\rho H_0^2)^2\right) \right] . \quad (23)$$

Now, by introducing the operator

$$O = [H_0^2, \rho] , \quad (24)$$

the function appearing in square brackets in Eq. (23) can be identified with

$$[2 \text{ tr}(\rho H_0^4 \rho) - 2 \text{ tr}\left((\rho H_0^2)^2\right)] = \text{tr}(O^\dagger O) , \quad (25)$$

which is the trace of a positive operator. Therefore, from Eq. (17) we have $\sigma > 0$, and hence

$$\partial_t S = 2 \sigma \text{ tr}(O^\dagger O) \geq 0 . \quad (26)$$

As a result, the linear entropy is a monotonically increasing function of time and, viceversa, the quantum state purity is monotonically decreasing. Asymptotically, the averaged density matrix will tend to the maximally mixed state proportional to the identity, and the off-diagonal coherences will be completely washed out. A zero growth rate for the linear entropy can occur only in the instance $[\rho, H_0] = 0$, which yields the trivial solution corresponding to the time-invariant density matrix.

As hinted above, in order to estimate the decoherence time it is convenient to work in the momentum representation. By means of this choice, we have to work with

$$\rho_{p,p'}(t) = \langle \mathbf{p} | \rho(t) | \mathbf{p}' \rangle . \quad (27)$$

By using the notation $E(p) = p^2/2m$, Eq. (19) becomes

$$\partial_t \rho_{p,p'} = \left[ -\frac{i}{\hbar}\left(E(p) - E(p')\right) - \sigma\left(E^2(p) - E^2(p')\right)^2 \right]\rho_{p,p'} , \quad (28)$$

whose solution is

$$\rho_{p,p'}(t) = \exp\left[-\frac{i(E(p) - E(p'))t}{\hbar} - \sigma\left(\Delta E^2\right)^2 t\right]\rho_{p,p'}(0) , \quad (29)$$

with $\Delta E^2 = \left(E^2(p) - E^2(p')\right)$. From Eq. (29), we observe that the time evolution conserves the diagonal elements whereas, as long as $\Delta E^2 \neq 0$ (i.e., no degeneracy), the off-diagonal terms of the density matrix decay exponentially, thereby realizing an effective localization in energy. The decay rate depends on both $\sigma$ and the fourth power of the characteristic energy scale of the system. From Eq. (29), we can then identify the decoherence time $\tau_D$:

$$\tau_D = \frac{1}{\sigma \left(\Delta E^2\right)^2} = \frac{\hbar^6}{16 m^2 \ell_p^4 t_p \left(\Delta E^2\right)^2} . \quad (30)$$

The decoherence time is strongly influenced by the actual energy regime: the larger the deviation from the Planck scale is, the longer $\tau_D$ becomes. We remark that Eq. (30) accounts for the size of the physical apparatus under consideration, as the squared

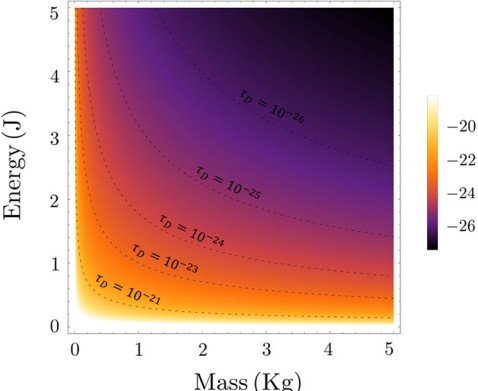

**Fig. 1 Behavior of $\log_{10}(\tau_D)$ as a function of the mass $m$ and the energy gap $\Delta E$.** As the level curves clearly show, the decoherence time $\tau_D$ is significantly reduced in the regions far from the microscopic scale and nearby the mesoscopic regime. Mass and energy are expressed in international units.

mass appears explicitly in the denominator of the expression for $\tau_D$. In Fig. 1, we draw the contour plot of the decoherence time as a function of the mass and energy scales.

Our investigation suggests that the effects stemming from quantum gravity induce an efficient bridge between the quantum and the classical domain at currently reachable energies. In order to verify this claim, it is important to quantify the quantum-gravitational decoherence times for typical classical and quantum systems at different scales. The physical dimensions and the corresponding $\tau_D$ for the examples considered are reported in Table 1.

As long as classical systems are concerned, the quantum-gravitational decoherence time is extremely short, and in one of the examples it is even close to the Planck time. This implies that quantum gravity could in principle be regarded as one of the main sources of the quantum-to-classical transition. Viceversa, when focusing on microscopic quantum systems, the results collected in Table 1 demonstrate that no coherence loss can be ascribed to the quantum nature of the gravitational interaction: the typical values for $\tau_D$ in the last three cases are far longer than the currently estimated age of the Universe. This occurrence leaves no room for a gravity-induced classical behavior in microscopic quantum systems. This is to be expected, as it would be paradoxical to observe a quantum-to-classical transition at the microscopic level induced by quantum effects.

As a final step, to make Eq. (30) more transparent, we can write the decoherence time as a function of the Planck energy $E_p = m_p c^2$ and the energy of the system in the rest frame $E = mc^2$. In so doing, we obtain

$$\tau_D = \frac{\hbar \, E_p^5}{16 \, E^2 \, \left(\Delta E^2\right)^2} . \quad (31)$$

In a similar fashion, also other decoherence times stemming from different physical settings[38,39,41,42] can be cast in the same form, which thus allows us to compare the predictions of our framework with those of the relevant gravitational decoherence models mentioned above. By focusing on the leading-order contributions only, we can build a table in which the main features of the different decoherence mechanisms are summarized.

By relying on the information contained in Table 2, we can pinpoint some interesting remarks. According to the above dependence of the various $\tau_D$ on the energy regime $E$, it appears that our model provides the shortest decoherence time above a

**Table 1 Values of $\tau_D$ for different physical systems.**

| Physical system | Energy scale (J) | Mass (Kg) | Value of $\tau_D$ (s) |
|---|---|---|---|
| Slow car | $2 \times 10^3$ | 900 | $2.749 \times 10^{-42}$ |
| Thrown tennis ball | 22.5 | 0.05 | $3.649 \times 10^{-26}$ |
| Cosmic dust | $3.125 \times 10^{-1}$ | $10^{-9}$ | $2.452 \times 10^{-3}$ |
| Benzene molecule | $4.981 \times 10^{-17}$ | $1.296 \times 10^{-25}$ | $2.263 \times 10^{92}$ |
| Neutron interferometer | $4.053 \times 10^{-21}$ | $1.675 \times 10^{-27}$ | $3.086 \times 10^{112}$ |
| Down quark at the quark epoch | $3.134 \times 10^{-14}$ | $8.592 \times 10^{-30}$ | $3.283 \times 10^{89}$ |

**Table 2 Comparison between different gravitational decoherence models.**

| Reference | Physical source of decoherence | Decoherence time |
|---|---|---|
| Breuer, Goklu and Lammerzahl[38] | Perturbation around flat spacetime | $\frac{\hbar E_p}{E^2}$ |
| Ellis, Mohanty and Nanopoulos[39] | Classical gravity near a wormhole | $\frac{\hbar E_p^3}{E^4}$ |
| Blencowe[41] | Thermal background of gravitons | $\frac{1}{k_B T}\frac{\hbar E_p^2}{E^2}$ |
| Anastopoulos and Hu[42] | Linearized gravity with thermal noise $\Theta$ | $\frac{1}{k_B \Theta}\frac{\hbar E_p^2}{E^2}$ |
| Current model | Fluctuating minimal length and deformation parameter | $\frac{\hbar E_p^5}{E^6}$ |

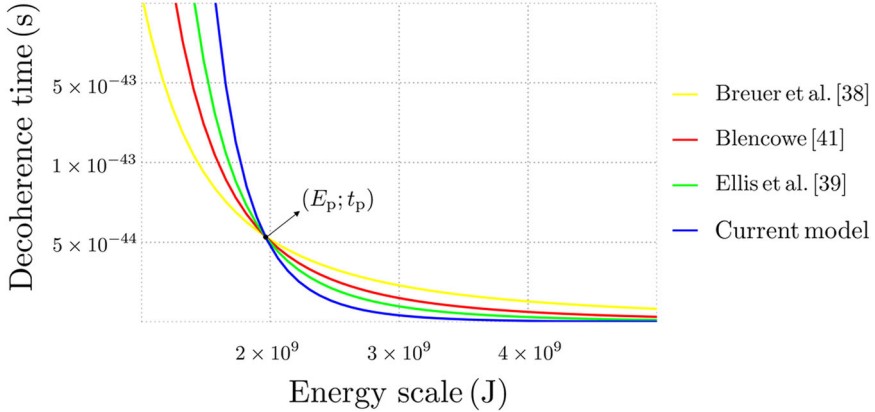

**Fig. 2 Behavior of the decoherence time as a function of the energy scale for different gravitational decoherence mechanisms.** Both quantities are expressed in international units. All curves intersect exactly at the Planck energy: $E = E_p$. For comparison purposes, for the model developed in ref. [41] we have required the thermal background of gravitons to yield a contribution compatible with the energy of the analyzed quantum system, that is $k_B T \simeq E$.

given energetic threshold (corresponding to a given mesoscopic scale), below which the scenario is completely reversed. As it might be expected, such an inversion occurs precisely at the Planck energy $E = E_p$, with the ensuing decoherence time given by $\tau_D = t_p$. This behavior is illustrated in Fig. 2, where we report $\tau_D$ as a function of $E$. For the time being, we can observe that, since our model predicts the longest decoherence time in the quantum regime (below the mesoscopic threshold fixed by the Planck scale), it appears to be the one best suited for experimental verifications with quantum coherent systems.

**Experimental verification.** Concerning the experimental verification by laboratory tests of any of the gravitational decoherence mechanisms proposed thus far, the canonical route in order to detect such effects is to perform matter-wave interferometry with massive particles in superposition states[45]. However, since the seminal experiment with fullerene[84] and other relatively light molecules, as of today no sufficiently heavy mesoscopic system has been successfully prepared in tabletop laboratory tests. On the other hand, despite this persistent limitation, some significant improvements have been achieved in recent years; indeed, in less

than a decade the largest molecule featuring quantum properties has moved from aggregates of <500 atoms[85] to structures made up of roughly 2000 atoms[86], thus paving the way for further progress in the hopefully not too distant future.

Other practically viable avenues to test gravitational decoherence models rely on quantum simulations with atom-optical platforms[87,88] and analogue models[89], which have been able to reproduce the spacetime curvature in the vicinity of black holes and wormholes.

Further potentially very efficient and promising schemes for the search of gravitational signatures in the decoherence processes are built upon quantum optical setups. The first proposal of a feasible experiment with a quantum optical system dates back to more then a decade ago[90,91]. According to these works, gravitational decoherence can be detected in near-Earth satellite tests involving entangled photons. The degradation of entanglement between the two photons is interpreted as a transfer of information from the bipartite quantum system to the environment via its coupling to each single photon, and the pervasive presence of gravity in empty interstellar space is regarded as the sole responsible for a

similar occurrence. For more details on this research field, see the comprehensive review[92]; for experimental developments and preliminary results, see ref. [93].

The same philosophy behind the quantum optical satellite probes is shared by laboratory tests of deformed commutation relations and other quantum gravity effects relying on systems of cavity optomechanics[94,95], with the further advantage that optomechanical setups allow for the implementation of universal schemes for decoherence detection. Specifically, in ref. [34] the authors have introduced a very general method to verify any gravitational decoherence model starting from a readout of the loss of entanglement between two prepared subsystems. This phenomenon can be revealed by means of either quantum tomography or (as we will see below) Clauser-Horne-Shimony-Holt (CHSH) correlation measurements[96]. According to ref. [34], a universal measure of decoherence is provided by means of the so-called min-entropy[97]. In a nutshell, from a physical point of view such an entropy accounts for the transfer (loss) of bipartite entanglement to a third party (an environment).

Formally, given a tripartite system made of distinguishable subsystems A, B and E, the degree of decoherence of A with respect to E is defined as[34]

$$\mathrm{Dec}(\mathrm{A}|\mathrm{E}) = \max_{\mathcal{R}_{\mathrm{E}\to\mathrm{B}}} F^2\left(\Phi_{\mathrm{AB}}, \mathbb{1}_{\mathrm{A}} \otimes \mathcal{R}_{\mathrm{E}\to\mathrm{B}}(\rho_{\mathrm{AE}})\right) , \quad (32)$$

where $\mathcal{R}_{\mathrm{E}\to\mathrm{B}}$ is the set of all quantum operations from E to B, $\rho_{\mathrm{AE}}$ is the reduced state of the bipartite system AE, i.e., the partial trace of the total density matrix $\rho_{\mathrm{ABE}}$ with respect to B (i.e., $\rho_{\mathrm{AE}} = \mathrm{Tr}_{\mathrm{B}}(\rho_{\mathrm{ABE}})$), $\Phi_{\mathrm{AB}}$ is a maximally entangled state on the bipartite system AB, $\mathbb{1}_{\mathrm{A}}$ is the unnormalized, fully decohered, maximally mixed state of subsystem A, and $F$ is the Uhlmann fidelity, that is[81]

$$F(\rho, \sigma) = \mathrm{Tr}\left(\sqrt{\sqrt{\rho}\, \sigma\, \sqrt{\rho}}\right) . \quad (33)$$

Systems A and B are the two parties which have been initially prepared in an entangled state, whilst E is the environment. If there is some mechanism that determines a loss of coherence, the entanglement between A and B is constantly degraded and at the same time the entanglement between A and E is enhanced. In the absence of decoherence the maximum of the state fidelity is realized for A and B in a maximally entangled state, while for a fully decohered system the maximum of the fidelity is realized for A in the fully decohered state $\mathbb{1}_{\mathrm{A}}/d_{\mathrm{A}}$, where $d_{\mathrm{A}}$ is the dimension of the Hilbert space associated to subsystem A. This quantification of decoherence via the fidelity interplay between maximally entangled and maximally decohered states is physically very transparent. Actually, it was shown in ref. [34] that the decoherence measure Dec(A|E) enjoys a remarkable relation to the min-entropy $H_{\min}(\mathrm{A}|\mathrm{E})_{\rho}$, that is the smallest Rényi conditional entropy providing a lower bound to the Shannon entropy of a statistical distribution $\rho$. Indeed, it turns out that[34,97]

$$H_{\min}(\mathrm{A}|\mathrm{E})_{\rho} = -\log_2\left(d_{\mathrm{A}} \max_{\mathcal{R}_{\mathrm{E}\to\mathrm{B}}} F^2\left(\Phi_{\mathrm{AB}}, \mathbb{1}_{\mathrm{A}} \otimes \mathcal{R}_{\mathrm{E}\to\mathrm{B}}(\rho_{\mathrm{AE}})\right)\right) = -\log_2\left(d_{\mathrm{A}}\mathrm{Dec}(\mathrm{A}|\mathrm{E})\right) . \quad (34)$$

The detailed proof of this remarkable relation between entropy and decoherence is thoroughly discussed in ref. [34].

Building on the above framework for the quantification of decoherence, we now discuss an optomechanical scheme for putting our model of quantum gravitational decoherence to the test. Let us consider two optomechanical systems, as shown in Fig. 3, where entangled photonic qubits are created in different conditions. In particular, one of the cavities possesses two fixed mirrors, while the other one is prepared so as to be subject to gravitational decoherence by allowing one of the mirrors to be

movable[34]. In both cavities, the mechanical oscillator is an atom or a molecular structure trapped in a harmonic potential and the two systems are prepared initially in an entangled state. By applying an external laser field, these oscillators jump from the ground state to an excited level; they then decay back to the ground state by emitting the cavity radiation that allows for an accurate study of their vibrational modes.

In a realistic environment, besides gravitational decoherence, we must necessarily take into account the phenomenon of mechanical heating. Hence, when estimating the mean phonon number $\bar{n}$ of the mechanical oscillator, for the cavity with a movable mirror we have to include the simultaneous concurrence of the two effects, which yields

$$\bar{n} = \frac{2\Lambda_{\mathrm{grav}}}{\gamma_{\mathrm{m}}} + \frac{2\Lambda_{\mathrm{heat}}}{\gamma_{\mathrm{m}}} , \quad (35)$$

where $\Lambda_{\mathrm{grav}} = 1/\tau_{\mathrm{D}}$, $\Lambda_{\mathrm{heat}} = k_{\mathrm{B}}T/\hbar Q$ and $\gamma_{\mathrm{m}} = \omega_{\mathrm{m}}/Q$ expresses the ratio between the frequency $\omega_{\mathrm{m}}$ of the harmonic potential and the cavity quality factor $Q$. It is then necessary to reach very low temperatures and very high quality factors in order for the second term to be suppressed and achieve the sensitivity required to detect the conjectured gravitational effect. The required best conditions then go as follows: $T = 10$ nK, $\omega_{\mathrm{m}} = 1\,\mathrm{s}^{-1}$, $\gamma_{\mathrm{m}} = 10^{-10}\,\mathrm{s}^{-1}$ (and consequently $Q = 10^{10}$). These are still very demanding conditions; in particular, the required operating temperature is still some orders of magnitude below the lowest temperature $T \simeq 0.1$ mK achievable with current cooling technologies of optomechanical systems[98,99].

Following the methods introduced in ref. [34], a rigorous evaluation of the decoherence measure Dec(A|E) yields

$$\mathrm{Dec}(\mathrm{A}|\mathrm{E}) = \frac{1}{4}\left\{1 + \sqrt{1 - \exp\left[-4(1 + 2\bar{n})\frac{g_0^2}{\omega_{\mathrm{m}}^2}\sin^2\left(\frac{\omega_{\mathrm{m}}t}{2}\right)\right]}\right\} , \quad (36)$$

where $g_0 = 1\,\mathrm{s}^{-1}$ is the single photon optomechanical coupling rate.

Recalling the expression of the decoherence time, Eq. (31), we see that in order to achieve a gravitational decoherence rate comparable to or larger than the one due to the mechanical heating process one needs large molecular structures with mass of the order $10^{-16}$ Kg (It is interesting to observe in passing that exactly this mass order of magnitude is the one required, in recent proposals for near-future tests of the quantum nature of the gravitational interaction[100,101]). It is foreseen that the use of large molecular structures with masses of the order $10^{-16}$ Kg will no longer represent an experimental limit in the next few years[100,101].

In the above experimental conditions, quantum gravitational decoherence effects become clearly detectable and distinguishable from other environmental decoherence mechanisms. In Fig. 4 we report the behavior of the decoherence rate Dec(A|E) as a function of time both in the presence and in the absence of quantum gravitational decoherence. As expected, in the former situation the loss of coherence is faster, as there are two concurring decoherence rates (gravitational and due to mechanical heating).

The full-fledged experimental verification of our quantum gravitational decoherence model lies in the possibility of detecting the quantum correlations of the photon sources inside the cavities via CHSH inequality-type measurements. The interplay between Bell's theorem, the CHSH inequalities and gravity has an interesting story per se and has been addressed in several different contexts, see for instance refs. [102–104]. For the present

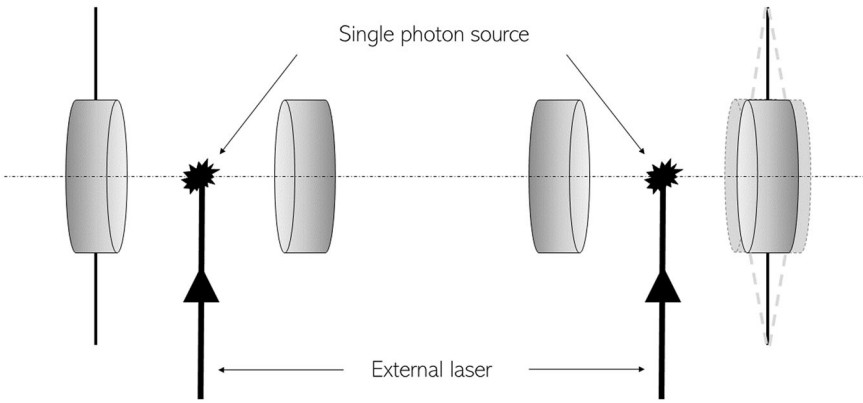

**Fig. 3 Optomechanical apparatus for the experimental test of quantum gravitational decoherence.** The left cavity has two fixed mirrors, whereas the right cavity contains a movable mirror. The same external laser excites two atomic (or molecular) systems that act as sources of entangled photons when they decay back to their respective ground states emitting in the process entangled photon radiation into the cavity.

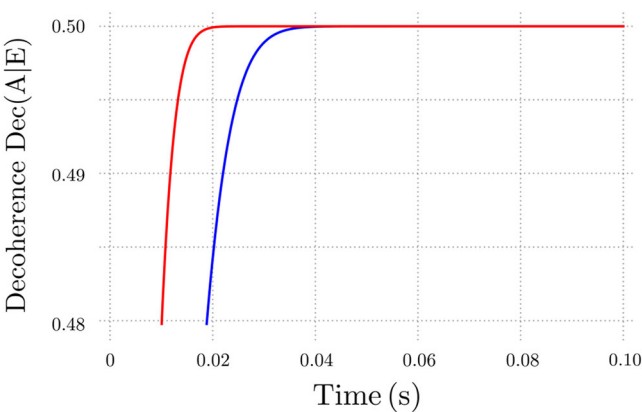

**Fig. 4 Behavior of the decoherence measure Dec(A|E) as a function of time.** We analyze the cases in which quantum gravitational effects are present (red line) and absent (blue line).

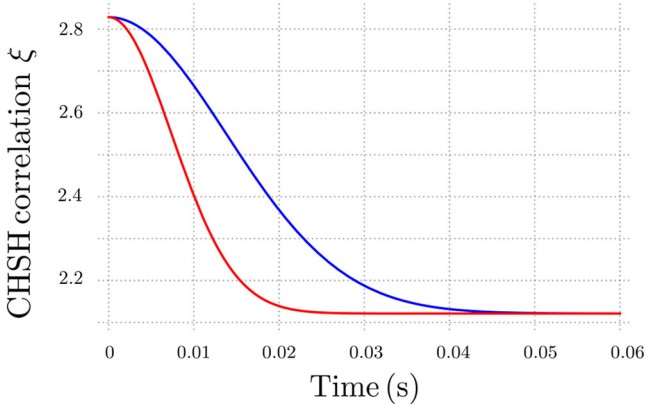

**Fig. 5 Behavior of the CHSH correlation $\xi(t)$ as a function of time.** We analyze the cases in which quantum gravitational effects are present (red line) and absent (blue line).

goal, the Bell-type CHSH correlation to be evaluated reads

$$\xi(t) = \text{tr}\left[\left(A_0 \otimes B_0 + A_0 \otimes B_1 + A_1 \otimes B_0 - A_1 \otimes B_1\right)\rho_{AE}\right] .$$
(37)

For the effective two-level systems under consideration the standard CHSH observables are

$$A_0 = \sigma_x , \qquad A_1 = \sigma_z , \qquad B_0 = \frac{\sigma_x - \sigma_z}{2} , \qquad B_1 = \frac{\sigma_x + \sigma_z}{2} ,$$
(38)

with $\sigma_x, \sigma_y, \sigma_z$ being the Pauli matrices.

As it can be deduced from Fig. 5, where we report the CHSH correlation $\xi(t)$ using the same parameter values adopted in Fig. 4, the presence or the absence of our quantum gravitational decoherence mechanism strongly discriminates between different experimental outcomes. Indeed, the function $\xi(t)$ for small (but accessible) times significantly departs from the value it would have if mechanical heating were the only source of decoherence. Therefore, the experimental scheme that we have illustrated appears suitable for the verification of our theoretical model.

## Discussion

We have investigated the decoherence process associated with the existence of a minimal length at the Planck scale and the corresponding deformed quantum uncertainty relations. Assuming that the minimal spatial scale of length and the ensuing deformation parameter $\beta$ are fixed only on average by space-time

quantum fluctuations and are thus fluctuating random quantities, we have derived a master equation for the averaged quantum density matrix, thus showing that such quantum open dynamics can be cast in a Lindblad form, which guarantees complete positivity and trace preserving. The dissipator in the master equation depends on $\beta$ in such a way to assure that the standard quantum mechanical dynamics is recovered in the limit $\beta \to 0$. The evolutions of the linear entropy and quantum state purity are monotonically increasing and monotonically decreasing in time, respectively, yielding diagonal output states asymptotically. By resorting to the momentum representation, we have estimated the decoherence time and we have evaluated typical values in order of magnitude for some physical systems of varying mass and energy scales.

We have also proposed an experimental setup with which to test our predictions via cavity optomechanics at low temperatures performing CHSH correlation measurements. In order to discriminate the quantum gravitational decoherence effects from those due to mechanical heating, we need heavy molecular oscillators and ultracold temperatures that are beyond the reach of current technologies but could become available in the near future[100,101].

In light of the above findings, we can conclude that the presence of a minimal length and a stochastic deformation parameter (as predicted by several leading candidates for a quantum theory of gravity) provides a valid decoherence mechanism capable of explaining some universal aspects of the quantum-to-classical

transition. By reasonably setting the mean deformation parameter at the constant value $\bar{\beta} = 1/\sqrt{t_{\rm p}}$, we have cleared our model from any dependence on free parameters that would require to be constrained a posteriori by experimental data, as long as the would-be free parameter $\alpha$ controlling the fluctuation intensity is set to be of order one.

The working hypothesis of a stochastic contribution to the deformation parameter $\beta$ is rooted in the dynamical scenarios for quantum gravitational effects[67] and the necessity to preserve the black hole complementarity principle. This ansatz leads to a transparent quantum-gravitational decoherence process that is quite immediate to discern from our formalism. For physical systems with mass and energy varying over a large spectrum, the values of the decoherence time $\tau_{\rm D}$ as summarized in Table 1 are entirely self-consistent and further corroborate the stochastic deformation picture. Furthermore, as reported in Fig. 2 and Table 2, comparison of our model with other relevant decoherence mechanisms proposed in the literature in recent years shows that our decoherence time is extremal, as it is the largest below the Planck scale and the smallest above it.

In summary, by assuming an emergent space-time equipped with a fluctuating fundamental length and a corresponding random deformation of the canonical commutation relations, our model yields a consistent and experimentally testable decoherence mechanism that explains the quantum-to-classical transition and provides a route for probing specific macroscopic consequences of the conjectured quantum nature of the gravitational interaction.

Concerning possible complementary and future developments, we recall that our quantum gravitational decoherence process occurs in energy space. On the other hand, the very same mechanism might arise also in connection with spatial localization. Indeed, the line of reasoning that we have carried out for the momentum-dependent GUP could be extended to include other features induced by quantum gravity, such as the so-called extended uncertainty principle, which accounts for the existence of a minimal momentum. Such a possibility was originally proposed in order to provide a foundation for the existence of the space-time curvature that becomes progressively more important at large distances[105]. The hypothesis of a minimal momentum scale is further supported by arguments based on the existence of a non-vanishing cosmological constant in the (anti)-de Sitter universe[106,107]. Following the same procedure developed in the present work, we plan to investigate the realization of a quantum-gravitationally induced spatial localization mechanism.

On a final note, we remark that we have studied the implications of a GUP that is in fact the lowest order in a series of corrections to the Heisenberg uncertainty relation. Consequently, our results hold true as long as the quantum gravitational effects are not exceedingly strong. Such an instance is always realized at the energy scales that can be probed in the current laboratory tests, and hence our predictions are suited for the foreseeable experimental verifications of a universal decoherence process.

On the other hand, it is conceptually worthwhile to work out a complete theory capable of taking into account all the possible higher-order extensions of the uncertainty principle[60,108–110]. The introduction of additional corrections to the Heisenberg uncertainty relation would plausibly modify and refine various quantitative traits; however, such corrections are likely to preserve the general qualitative features outlined in the present work. We plan to tackle these issues in upcoming follow-on investigations.

## Data availability
Data sharing not applicable to this article as no datasets were generated or analyzed during the current study.

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

## Acknowledgements

This work is supported by MUR (Ministero dell'Università e della Ricerca) under project PRIN 2017 "Taming complexity via QUantum Strategies: a Hybrid Integrated Photonic approach" (QUSHIP) Id. 2017SRNBRK. We thank V. Bittencourt for helpful discussions.

## Author contributions

L.P. and F.I. conceived the project. L.P. performed calculations and drafted the paper. F.I. supervised the work. Both authors discussed the results and edited the paper.

## Competing interests

The authors declare no competing interests.
