## [Peer Review File · Nature Communications]

Reviewers' Comments:

Reviewer #1:

Remarks to the Author:

This manuscript presents new theoretical results that establish a connection between a candidate Planck length scale modified Heisenberg uncertainty principle and an apparent fundamental decoherence mechanism in a particle system's kinetic energy eigenstate basis. The work is timely, given the existence of several recent theoretical investigations on the possible role of (quantum) gravity in enforcing classicality. Furthermore, the authors find a surprisingly short decoherence time as system mass/energy scales extend into the mesoscopic range and beyond. In my opinion, the manuscript is therefore potentially worthy of publication in Nature Communications, provided the authors satisfactorily consider the following points through added discussion, equations and citations where relevant.

1) Note typo: "pioneerint"->pioneering.

2) Equation (19) on the face of it appears a little unnatural, in particular the 'high' powers of H_0 in the dephasing term. Note, that similar looking equations have been derived through related considerations. For example, Milburn [Phys. Rev. A 44, 5401 (1991)] assumes a fundamental minimum time scale [see also J. Ellis et al., Phys. Lett. B 221, 113 (1989)], which results in a similar looking dephasing term, but with a lower power in H (full Hamiltonian/system energy). In light of this, is there a compelling reason to focus on Planck length scale vs Planck time scale corrections to the Schrödinger equation? It seems a bit odd that there would be such a qualitative difference in the resulting master equations which may be a consequence of mixing non-relativistic QM with quantum relativistic spacetime geometry considerations. If it were instead possible to work with a relativistic quantum field system with speculated Planck length/time-modified field operator commutation relations, would the resulting master equation be qualitatively different from (19)?

3) How can (19) be understood in terms of genuine decoherence? From the derivation of (19), there does not appear to be a naturally identified environment with which the system becomes entangled, contrary to the early discussion in the introduction. Please note the works e.g. by Blencowe, Phys. Rev. Lett. 111, 021302 (2013), Anastopoulos and Hu, Class. Quantum Grav. 30 165007 (2013), and Oniga and Wang Phys. Rev. D 93, 044027 (2016), where genuine decoherence by stochastic gravitational environments is considered. The latter works predict decoherence in the energy basis similar to Milburn's model and the present manuscript.

4) Equation (30) might look more elegant/clear if cast alternatively in terms of e.g., the Planck energy, the system rest mass energy, ΔE and Planck's constant \hbar . It would then be interesting to compare/contrast the numerical predictions arising e.g., from the above decoherence references. With the larger power of Planck energy and ΔE appearing in (30) vs the above decoherence reference predictions, it would seem that (30) gives increasingly shorter decoherence times above a certain mesoscopic mass/energy scale, but increasing longer times below this scale compared to the predictions from the above references.

Miles Blencowe

Reviewer #2:

Remarks to the Author:

In the present manuscript, the authors consider the influence of a minimal measurable length in the quantum-to-classical transition. Specifically, after finding the corresponding master equation, the authors study the evolution of entropy of a quantum system and the decoherence time. Such description is considered starting from a widely known model, which goes under the name of Generalized Uncertainty Principle (GUP) and is introduced in Eqs.(1-3).

To my knowledge, it is the first time that such analysis has been carried out. Furthermore, regarding the modification parameter as a dynamical quantity, rather than static as it is usually

considered, is an interesting point.

I mainly have two concerns:

1. In Eq.(5), an approximation of the high-energy quantities in terms of the low-energy ones is proposed. Although such approximation is generally considered in the literature, it presents some problems. Specifically, as it can be seen in arXiv:2005.12258 [gr-qc], the operator \hat{x}_j is not symmetric. Although such point does not affect the results of the paper, at least as far as I can see, it would be useful if the authors could comment about it.

2. My second concern regards the dynamical nature of the deformation parameter. In particular, in the context of a deformed commutation relation between position and momentum, e.g., as in Ref.[66], values $\beta \leq 0$ represent problematic cases. In fact, $\beta=0$ would mean the absence of any modification with respect to the Heisenberg principle. Furthermore, a negative value for β would break many of the relations in Ref.[66], among which the definition of the minimal length itself.

Thus, I would recommend the authors to comment on the points above before the manuscript be accepted for publication.

Reviewer #3:

Remarks to the Author:

Referee Report: "Quantum gravitational decoherence"

The article deals with a very important open problem in Physics: how to justify the quantum-to-classical transition, i.e. how the classical world of definite events emerges from the quantum world where events seem to fade away. The idea the authors pursue, is that quantum gravity can provide an answer to the problem.

Quantum gravity models predict a minimal length at the Planck scale, one of whose consequences is a modification of the Heisenberg commutation relations between position and momentum. The authors then take a minimal form of the modified commutation relations, and add a stochastic component to the deformation parameter controlling the magnitude of the modifications. The associated Schrödinger equation becomes stochastic, leading to a Lindblad equation for the density matrix, to lowest order. From there, entropy and purity variation are computed, as well as the decoherence time. Numerical estimates are provided, showing that the decoherence time decreases when moving from the micro- to the macro-world, thus potentially explaining the quantum-to-classical transition.

The work is interesting but not groundbreaking: the idea that (quantum) gravity might explain the quantum-to-classical transition is not new. There already exist proposals of gravitational decoherence in the literature; one such an example is [30] (where the issue of the quantum-to-classical transition is explicitly addressed), others are reviewed in [26]. That considered by the authors in the present article adds to those, without changing the overall picture.

It is not true that the model here considered is without free parameters, which is one of the strong claims of the authors: In Eq. (9) the authors set the strength of the correlation function of the noise equal to the square root of the Planck time. This is an arbitrary choice, since the correlator is assumed and not derived. Therefore, as it stands, the time correlator is a free parameter. [Also the GRW models and the DP model, quoted by the authors in the introduction, can be made parameter-free by fixing the numerical values of the parameters in some way.]

After Eq. (19) the authors write "From the above analysis, we can conclude that quantum gravity does indeed affect quantum coherence due to the existence of a minimal length at the Planck scale". In the conclusions they add "In light of the above findings, we can conclude that the presence of a minimal length ... furnishes a valid decoherence mechanism capable of explaining some universal aspects of the quantum-to-classical transition". These statements are too strong

and misleading. It is not the presence of a minimal length that induces decoherence, it is the assumption that the deformation parameter is stochastic. This is not a secondary detail, because the authors do not provide a strong motivation as to why it should be stochastic, other than the fact that different models predict different values for it. To make an analogy, I can imagine that different models exist, predicating (slightly) different ages of the universe, but that does not imply that the age of the universe is stochastic.

Another weak point is that all calculations after Eq. (6) are conventional and not particularly enlightening. It is clear to anyone with a basic knowledge of the theory of open quantum systems that the master equation (19), which follows from (6) quite straightforwardly, implies an increase of entropy and a decrease of purity, and also a decoherence time as given by Eq. (30).

It would have been interesting if the authors had performed a comparison between the different types of gravitational decoherence (theirs against those already discussed in the literature) to decide which one is stronger and in which situations. It would have been even more interesting if the authors had proposed an experiment which could potentially detect the quantum gravitational decoherence they present, by establishing the working conditions required to measure it: mass of the system, type of measurement, pressure and temperature to lower environmental noises, ...

To conclude, the analysis seems correct but does not add much to the field, and further work and comparisons with other results in the literature should be carried out, before considering it for publication.

Minor points.

1. In listing the reviews on decoherence and open quantum systems [6-10], quoting Schlosshauser 3 times out of 5 is disproportionate; the field is very vast and other important references, for example the book of Breuer and Petruccione, are not even mentioned.
2. The long paragraph in the Introduction on spontaneous wave function collapse models denotes a very poor knowledge of the field. Just as an example, referring to [24, 25] for "for important developments in GRW-like models" is surprising, since [24] is from 2000 and [25] has nothing to do with GRW. Are the authors really sure that nothing important occurred in the field over the last 20 years?
3. The GRW model is defined in terms of two parameters, not one as stated by the authors.

Dear Editors,

We are grateful to the Reviewers for their insightful comments and recommendations, which helped us to improve our manuscript. In compliance with their requests, in what follows we list all the amendments/additions we have made to the original draft, point by point for each Reviewer. All corrections and additions in the revised manuscript appear in red color text in the pdf version.

REVIEWER #1

Comment 1) "Note typo: "pioneerint" -> pioneering."

Reply: We have corrected the typo and changed "pioneerint" into "pioneering".

Comment 2) "Equation (19) on the face of it appears a little unnatural, in particular the 'high' powers of H_0 in the dephasing term. Note, that similar looking equations have been derived through related considerations. For example, Milburn [Phys. Rev. A 44, 5401 (1991)] assumes a fundamental minimum time scale [see also J. Ellis et al., Phys. Lett. B 221, 113 (1989)], which results in a similar looking dephasing term, but with a lower power in H (full Hamiltonian/system energy). In light of this, is there a compelling reason to focus on Planck length scale vs Planck time scale corrections to the Schrödinger equation? It seems a bit odd that there would be such a qualitative difference in the resulting master equations which may be a consequence of mixing non-relativistic QM with quantum relativistic spacetime geometry considerations. If it were instead possible to work with a relativistic quantum field system with speculated Planck length/time-modified field operator commutation relations, would the resulting master equation be qualitatively different from (19)?"

Reply: In reply to this comment, we have added a new paragraph following Eq. (19) on page 5 of the revised manuscript, explaining that the focus on the Planck length scale is, more than a choice, a consequence motivated from string theory as well as from other quantum gravitational models. In fact, a fully relativistic treatment would involve the introduction of a much more complex generalized uncertainty principle, which in turn would entail a behavior that deviates from the one currently investigated. Since the present form of the GUP is the one currently well-established, we have decided to work with Eq. (1) rather than with other generalizations that would be in principle possible. Moreover, as pointed out in the revised draft, a fully relativistic treatment would introduce several difficult and yet partially unresolved issues, not least a notion of decoherence that applies to fields rather than single-particle states. Having to deal with fields, one might try to proceed with the definition of a suitable effective action and from there build the relevant decoherent dynamics for the reduced system. Let alone the formidable technical difficulties facing it, such research program is at the moment not sufficiently well developed to allow a comprehensive theoretical analysis.

Comment 3) "How can (19) be understood in terms of genuine decoherence? From the derivation of (19), there does not appear to be a naturally identified environment with which the system becomes entangled, contrary to the early discussion in the introduction. Please note the works e.g. by Blencowe, Phys. Rev. Lett. 111, 021302 (2013), Anastopoulos and Hu, Class. Quantum Grav. 30 165007 (2013), and Oniga and Wang Phys. Rev. D 93, 044027 (2016), where genuine decoherence by stochastic gravitational environments is

considered. The latter works predict decoherence in the energy basis similar to Milburn's model and the present manuscript."

Reply: As remarked and better clarified in an added text paragraph before Eq. (2) on page 2 of the revised manuscript, we are actually studying a system in a gravitational environment; the differences with other approaches lies in the basic aspects that we have chosen to address as well as the way such information is encoded in the investigation. Our framework rest on two fundamental non-perturbative features of quantum gravity: a minimal length and a fluctuating spacetime. These are in turn directly incorporated in the canonical commutation relations. The ensuing decoherent dynamics is thus essentially caused by the fast dephasing associated to the fluctuating deformation parameter β .

Comment 4) "Equation (30) might look more elegant/clear if cast alternatively in terms of e.g., the Planck energy, the system rest mass energy, ΔE and Planck's constant \hbar . It would then be interesting to compare/contrast the numerical predictions arising e.g., from the above decoherence references. With the larger power of Planck energy and ΔE appearing in (30) vs the above decoherence reference predictions, it would seems that (30) gives increasingly shorter decoherence times above a certain mesoscopic mass/energy scale, but increasing longer times below this scale compared to the predictions from the above references."

Reply: We thank the Reviewer for having drawn this important point to our attention. To comply with her/his request, we have derived and added the new Eq. (31) on pages 7-8 of the revised manuscript, together with an extended discussion as well as a further table (Table II on page 8 of the revised manuscript) and a further figure (Figure II on page 9 of the revised manuscript) in which, as suggested by the Reviewer, we review and compare with our model other different approaches to gravitational decoherence. This supplementary analysis indicates that the insight of the Reviewer is indeed correct: among all the gravitational decoherence mechanisms, our decoherence time is extremal, being the longest below the Planck energy threshold and the shortest above it.

REVIEWER #2

Comment 1) "In Eq.(5), an approximation of the high-energy quantities in terms of the low-energy ones is proposed. Although such approximation is generally considered in the literature, it presents some problems. Specifically, as it can be seen in arXiv:2005.12258 [gr-qc], the operator \hat{x}_j is not symmetric. Although such point does not affect the results of the paper, at least as far as I can see, it would be useful if the authors could comment about it."

Reply: By virtue of the Reviewer's observation, we have improved our considerations revolving around the peculiar nature of the position operator in the present framework (see new text paragraph added after Eq. (5) on page 3 of the revised manuscript and the relevant references quoted).

Comment 2) "My second concern regards the dynamical nature of the deformation parameter. In particular, in the context of a deformed commutation relation between position and momentum, e.g., as in Ref.[66], values $\beta \leq 0$ represent problematic cases. In fact, $\beta=0$ would mean the absence of any modification with respect to the Heisenberg principle. Furthermore, a negative value for β would break many of the relations in Ref.[66], among which the definition of the minimal length itself.

Reply: We have added a footnote to page 3 of the revised manuscript to convey the main differences with the case of positive β and provide an answer to the Reviewer's observation. In fact, there is a

substantial body of recent literature pointing to the fact that β may indeed be negative and discussing the implications arising from such a scenario. On this basis, we review the arguments on the viability of a negative deformation parameter. These arguments further help to understand how β can effectively be regarded as a fluctuating dynamical quantity.

REVIEWER #3

Central remark part 1) "It is not true that the model here considered is without free parameters, which is one of the strong claims of the authors: In Eq. (9) the authors set the strength of the correlation function of the noise equal to the square root of the Planck time. This is an arbitrary choice, since the correlator is assumed and not derived. Therefore, as it stands, the time correlator is a free parameter. [Also the GRW models and the DP model, quoted by the authors in the introduction, can be made parameter-free by fixing the numerical values of the parameters in some way.]"

Reply: We have revised and expanded the text following Eq. (9) on page 4 of the revised manuscript concerning the assumption of a fluctuating deformation parameter β and a reasonable approach to the fixing of its constant mean. Given the current consensus in the literature on β being of order unity, we fix its mean value at one, so that no subsequent arbitrary choice needs to be imposed.

Central remark part 2) "After Eq. (19) the authors write "From the above analysis, we can conclude that quantum gravity does indeed affect quantum coherence due to the existence of a minimal length at the Planck scale". In the conclusions they add "In light of the above findings, we can conclude that the presence of a minimal length ... furnishes a valid decoherence mechanism capable of explaining some universal aspects of the quantum-to-classical transition". These statements are too strong and misleading. It is not the presence of a minimal length that induces decoherence, it is the assumption that the deformation parameter is stochastic. This is not a secondary detail, because the authors do not provide a strong motivation as to why it should be stochastic, other than the fact that different models predict different values for it. To make an analogy, I can imagine that different models exist, predicating (slightly) different ages of the universe, but that does not imply that the age of the universe is stochastic."

Reply: We have revised and expanded the text reviewing the arguments in support of a dynamical nature of β that stem from the hypothesis of a fluctuating spacetime at the Planck scale, as predicted (among others) by the quantum foam scenarios and by loop quantum gravity. Additionally, we have further stressed that the source of decoherence is crucially due to the fluctuating nature of the deformation parameter.

Central remark part 3) "It would have been interesting if the authors had performed a comparison between the different types of gravitational decoherence (theirs against those already discussed in the literature) to decide which one is stronger and in which situations."

Reply: Prompted by this comment, on pages 7-9 of the revised manuscript we compare different gravitational decoherence mechanisms in detail and we identify the energy regimes where our predictions significantly stand out with respect to those of the other models, as shown in the new Table II on page 8 and the new Figure 2 on page 9 of the revised manuscript, and discussion thereafter. As already mentioned in point 4) of the reply to Reviewer # 1, it turns out that among all the gravitational decoherence mechanisms being compared, our one appears to be extremal, in the sense that our decoherence time is the longest below the Planck energy threshold and the shortest above it.

Central remark part 4) "It would have been even more interesting if the authors had proposed an experiment which could potentially detect the quantum gravitational decoherence they present, by establishing the working conditions required to measure it: mass of the system, type of measurement, pressure and temperature to lower environmental noises, ..."

Reply: We have added a discussion on pages 7-8 of the revised manuscript concerning possible experimental tests. We have emphasized how, at present, the only promising way to probe hypothetical mechanisms of gravitational decoherence (both classical and quantum) is to rely on matter-wave interferometry with massive particles in superposition states. However, since the largest molecular aggregates that exhibit quantum features are far lighter than the supposed systems with which an explicit experimental test could be feasible within the current technological scopes and limits, the only alternative method that we feel we can suggest in principle for future experimental verifications of any quantum gravitational decoherence mechanism (including our approach) might be provided by quantum simulators with atom-optical platforms, in analogy with what has been done concerning sonic analogues of quantum black holes. On a general note, we should stress that the experimental addressing of gravitational decoherence is extremely challenging in all the frameworks proposed so far in the literature. On the other hand, at the current pace, the experimental progress gives hope for significant improvements in the not so far future.

Minor point 1) "In listing the reviews on decoherence and open quantum systems [6-10], quoting Schlosshauser 3 times out of 5 is disproportionate; the field is very vast and other important references, for example the book of Breuer and Petruccione, are not even mentioned."

Reply: We thank the Reviewer for this observation; following her/his advice, we have properly extended and updated the bibliography including recent relevant references.

Minor point 2) "The long paragraph in the Introduction on spontaneous wave function collapse models denotes a very poor knowledge of the field. Just as an example, referring to [24, 25] for "for important developments in GRW-like models" is surprising, since [24] is from 2000 and [25] has nothing to do with GRW. Are the authors really sure that nothing important occurred in the field over the last 20 years?"

Reply: We agree with the Reviewer on this point as well and we have thoroughly searched, expanded and updated the relevant references.

Minor point 3) "The GRW model is defined in terms of two parameters, not one as stated by the authors."

Reply: We thank the Reviewer for having drawn this inaccuracy to our attention; this aspect has been properly amended in the revised manuscript.

We are confident that, in the present revised and improved form, the paper may now be suitable for publication in Nature Communications.

Sincerely yours,

Fabrizio Illuminati,

Luciano Petruzziello

Reviewers' Comments:

Reviewer #1:

Remarks to the Author:

I have carefully read the authors' replies to my and the other referees' comments, as well as their revised manuscript. I feel that they have satisfactorily addressed the comments and therefore recommend publication in Nature Communications.

Reviewer #2:

Remarks to the Author:

I think that the Authors have satisfactorily responded the comments from the Referees. Specifically, the Authors expanded on the nature of the parameter and its stochastic behavior. Although the interplay between quantum gravity and decoherence has been considered in the past, to my knowledge this is the first time that a modified uncertainty relation is considered with a stochastic flavor and studied in correlation to decoherence. I thus think that the manuscript is worth of publication in Nature Communications.

Reviewer #3:

Remarks to the Author:

Report on "Quantum gravitational decoherence"

I thank the authors for the reply to my previous report. However, I do not think they addressed my criticisms, while others I made were ignored.

First. The logic of the paper seems to be the following:

- A. Several Quantum Gravity models predict a minimal length at the Planck scale, which implies a modified Heisenberg uncertainty principle (Eq. (1)).
- B. Modifications of the Heisenberg uncertainty principle are taken as a signature of gravitational decoherence (new red text at the beginning of Section II).

This is a bold assumption, whose justification I fail to grasp. Take for example standard QED: it predicts "photonic decoherence" on matter (when tracing out the electromagnetic degrees of freedom), but this cannot be read off directly from the commutation relations. Then, when dealing with quantum gravity, why should the additional terms appearing in the Heisenberg uncertainty principle be linked directly to gravitational decoherence? This assumption needs a justification, while in the paper it is simply stated.

Second. Accepting that the additional terms appearing in the Heisenberg uncertainty principle can be taken as a signature of gravitational decoherence, the next assumption is that the parameter beta is a fluctuating quantity. At this regard, in their reply to my previous comment, the authors write "...dynamical nature of beta that stem from the hypothesis of a fluctuating spacetime at the Planck scale". I do not understand the reasoning.

Let us take again the example of QED; also in this case, there are vacuum fluctuations (which have observable consequences), but as far as I know they do not lead to a stochastic charge or stochastic mass. Vacuum fluctuations do lead to decoherence, but this is a different story, and is treated in a different manner, with charge and mass having well defined values.

Third. Accepting that beta is random, one has to decide on its statistical properties. It is reasonable to assume that it is a Gaussian white noise with a fixed mean. Then the noise is fully characterized by the mean and the correlation. In particular, in Eq. (9) the correlation is taken equal to the Planck time. Why this choice? [In their reply to my comment at this regard (n. 1 according to the author's list), the authors fail to answer the question. I am fine with taking the mean equal to unity, as the authors write in the reply; my question was about the correlator.]

Making again reference to QED, one could equally well say that vacuum fluctuations induce "photonic decoherence" on electrons which shows up as a white noise with correlation time equal to $[e^2 / 4 \pi \epsilon_0 m_e c^3]$, with obvious meaning of the constants (I hope I computed the dimensions correctly). I made this time up now, and one could consider it as the electromagnetic analogous of the Planck time. It is clear that those constants will appear somewhere in the calculations, but the decoherence effect coming from the electromagnetic degrees of freedom is much more complex – see for example the book "Relativistic Quantum Measurement and Decoherence" by Breuer and Petruccione.

As such, the assumption that the beta's fluctuations are proportional to the Planck time is an assumption. In this sense I restate my claim that the model is not parameter-free. There is a free parameter, which has been set equal to the Planck time, without an a-priori justification.

In summary, the assumptions which lead to Eq. (10) are not justified.

Coming to my other criticisms commented by the authors, I thank the authors for adding table II with a comparison with other models of gravitational decoherence, which have been proposed in the literature (n. 3 according to the author's list).

The discussion about possible ways of measuring it (n. 4 according to the author's list) however remains rather poor. The authors claim "at present the only practically viable avenue to test gravitational decoherence models relies on quantum simulations with atom-optical platforms". Again, this is a bold claim, which dismisses entirely other possible platforms for testing gravitational decoherence, such as opto-mechanics, or space experiments with photons. How do the authors substantiate their claim? Clearly, a full analysis requires a lot of work, but some rule-of-thumb estimates should not be difficult to give.

For the above reasons, I cannot recommend the paper for publication.

Dear Editors,

We would like to thank you for giving us the opportunity to submit a revised draft of our manuscript a second time. We would like to thank Reviewers #1 and #2 for their very useful remarks and their positive assessment of our work.

We also thank Reviewer #3 whose sharp critical observations helped us to greatly improve both the original manuscript as well as the two further revisions.

Indeed, prompted by Reviewer # 3 further stimulus, in this second revision, throughout the manuscript we have better clarified the logic that motivates our proposal for quantum gravitational decoherence, and we have added an entirely new Section, Section V, on the experimental verification of our model with optomechanical setups and ultracold massive molecular structures.

We have provided a detailed description of the experimental scheme and of its functioning, as well as precise predictions on the time growth of the decoherence rate and of the time decay of the nonlocal quantum correlations. Moreover, the predictions are experimentally testable in a range of physical parameters of the cavity optomechanical setups that are not impossibly distant (about 3-4 orders of magnitude) from the ones currently achievable in the laboratory.

In summary, we believe that in this second revised resubmission we have now produced a consistent model that assumes an emergent spacetime equipped with a fluctuating fundamental length scale fixed on average. In turn, this hypothesis implies a random Planck-scale deformation of the canonical commutation relation and of the uncertainty principle. The model then yields a consistent and experimentally testable decoherence mechanism for the quantum-to-classical transition, as well as a route for probing specific macroscopic consequences of the conjectured quantum nature of the gravitational interaction.

To us, the work is now thorough and complete, as we have explored all aspects that were left out in the original submission and that we have covered in the subsequent revisions thanks to the many valuable comments by the anonymous Reviewers.

In the following we give a detailed reply to the second round of comments by Reviewer # 3:

Comment 1) *“First. The logic of the paper seems to be the following:*

A. Several Quantum Gravity models predict a minimal length at the Planck scale, which implies a modified Heisenberg uncertainty principle (Eq. (1)).

B. Modifications of the Heisenberg uncertainty principle are taken as a signature of gravitational decoherence (new red text at the beginning of Section II).

This is a bold assumption, whose justification I fail to grasp. Take for example standard QED: it predicts “photonic decoherence” on matter (when tracing out the electromagnetic degrees of freedom), but this cannot be read off directly from the commutation relations. Then, when dealing with quantum gravity, why should the additional terms appearing in the Heisenberg uncertainty principle be linked directly to gravitational decoherence? This assumption needs a justification, while in the paper it is simply stated.”

Reply to Comment 1): We thank the Reviewer for this comment that helped us to further clarify a point that was not very clearly stated in the previous versions of our manuscript, and to further expand on the logic of the paper.

Our starting point are the modified canonical commutation relations (modified CCRs) as predicted heuristically by several models of quantum gravity. These deformed CCRs depend on a deformation parameter, β , that is heuristically derived from and depends on the assumption of a granular dimension of space that should set in at the Planck scale. Indeed, in the non-relativistic limit, the modified CCRs induce a modified momentum operator to first order in the deformation parameter β . As momentum is conjugate to position, this modification of the momentum, i.e. of the generator of the space translations, is consistent only as long as space is assumed discretized precisely according to the modified CCRs themselves (other more exotic modifications of the CCRs due to other forms of discretization are possible, and we briefly comment on them in the conclusions).

The modification in the momentum operator in turn implies a significant modification in the Schrödinger dynamics for the state vector, with the introduction of an extra Hamiltonian term that is dependent on the deformation parameter β and vanishes monotonically in the limit of a vanishing β (see Section II “Generalized Uncertainty Principle And Modified Schrödinger Dynamics” of the revised manuscript).

Up to this point we do not take the dependence on β and the additional terms appearing in the Heisenberg uncertainty principle as a signature of gravitational decoherence, but only as a source of a modified unitary quantum dynamics. Decoherence sets in only as the Schrödinger dynamics acquires a stochastic character, and this occurs if the β parameter is a fluctuating quantity inducing a stochastic Schrödinger equation.

Comment 2) *“Second. Accepting that the additional terms appearing in the Heisenberg uncertainty principle can be taken as a signature of gravitational decoherence, the next assumption is that the parameter β is a fluctuating quantity. At this regard, in their reply to my previous comment, the authors write “...dynamical nature of β that stem from the hypothesis of a fluctuating spacetime at the Planck scale”. I do not understand the reasoning.*

Let us take again the example of QED; also in this case, there are vacuum fluctuations (which have observable consequences), but as far as I know they do not lead to a stochastic charge or stochastic mass. Vacuum fluctuations do lead to decoherence, but this is a different story, and is treated in a different manner, with charge and mass having well defined values.”

Reply to Comment 2): Following the reply to comment 1), and the second revised version of the manuscript, where we have clarified that the additional terms in the CCRs are not yet *per sé* a source/signature of gravitational decoherence, but only induce a modified Schrödinger dynamics for the state vector or Liouville-von Neumann unitary evolution for the state projector, in Section III of our work we discuss how a fluctuating deformation parameter β leads to a convex statistical ensemble of state projectors and therefore to a non-unitary dynamics for the averaged density matrix that in turn realizes the quantum gravitationally induced decoherence. The crucial difference with QED and vacuum fluctuations of ordinary quantum field theory is that in the context of quantum foam models of quantum gravity is the minimal quantum of space dimension itself that fluctuates around an average value fixed at the Planck scale. In ordinary quantum field theory and in standard quantum mechanics the quantum of action \hbar is taken to be a fixed quantity that does not fluctuate and thus it cannot be *per sé* a source or signature of any diffusion or decoherence process.

Of course, at this point the task we are left with is to provide a sound physical picture on the possible origin of the assumption that the minimal space dimension and the related deformation parameter for the space translations are fluctuating quantities fixed only in the mean.

It is clear that given the current lack of a consistent theory of quantum gravity, it is impossible to be certain what spacetime would look like at small scales. Assuming a quantum nature of the gravitational interaction, it has been repeatedly suggested in the literature that spacetime itself might consist of many small, ever-changing regions ("bubbles") in which space and time are not definite, but fluctuate in a foam-like manner.

Clearly, these hypothetical "bubbles" would be in general different stochastic realizations, each with a different spatial extensions; therefore the length dimension of this hypothetical foam is by definition a fluctuating quantity, **giving rise to a definite minimal scale of length only in the sense of an average**. The hypothetical random nature of the minimal length then induces also a deformation parameter β that is a fluctuating quantity itself.

Of course this is just an hypothesis, but a sound and legitimate one; and, if we may add, a very fascinating one. In this second revision we have further re-formulated the hypothesis in its clearest possible terms in several places, from the abstract through the introduction to the conclusions (all new and modified lines of text are now in red color).

Comment 3) *"Third. Accepting that beta is random, one has to decide on its statistical properties. It is reasonable to assume that it is a Gaussian white noise with a fixed mean. Then the noise is fully characterized by the mean and the correlation. In particular, in Eq. (9) the correlation is taken equal to the Planck time. Why this choice? [In their reply to my comment at this regard (n. 1 according to the author's list), the authors fail to answer the question. I am fine with taking the mean equal to unity, as the authors write in the reply; my question was about the correlator.]"*

Making again reference to QED, one could equally well say that vacuum fluctuations induce "photonic decoherence" on electrons which shows up as a white noise with correlation time equal to $[e^2 / 4 \pi \epsilon_0 m_e c^3]$, with obvious meaning of the constants (I hope I computed the dimensions correctly). I made this time up now, and one could consider it as the electromagnetic analogous of the Planck time. It is clear that those constants will appear somewhere in the calculations, but the decoherence effect coming from the electromagnetic degrees of freedom is much more complex – see for example the book "Relativistic Quantum Measurement and Decoherence" by Breuer and Petruccione.

As such, the assumption that the beta's fluctuations are proportional to the Planck time is an assumption. In this sense I restate my claim that the model is not parameter-free. There is a free parameter, which has been set equal to the Planck time, without an a-priori justification."

Reply to Comment 3): We beg to politely disagree with the Reviewer on the choice of the correlator being without any a-priori justification, thus making the model dependent on a completely free parameter. The point is that as the quantum gravitational effects become relevant at the Planck scale, we cannot think of any other better time-dimensional candidate than the Planck time to control the intensity of the fluctuations. It is the most natural choice under the assumption that quantum gravity sets in precisely at such a scale.

However, to allow for an alternate definition as recommended by Reviewer #3, we have also considered the case in which the intensity of the fluctuations can be taken as proportional to t_p , thus making the proportionality factor the would-be free parameter of the model. Otherwise, the proportionality factor is taken of order unity throughout in the paper. Taking either of the two alternatives leads of course exactly to the same physical results.

Comment 4) *“Coming to my other criticisms commented by the authors, I thank the authors for adding table II with a comparison with other models of gravitational decoherence, which have been proposed in the literature (n. 3 according to the author’s list).*

The discussion about possible ways of measuring it (n. 4 according to the author’s list) however remains rather poor. The authors claim “at present the only practically viable avenue to test gravitational decoherence models relies on quantum simulations with atom-optical platforms”. Again, this is a bold claim, which dismisses entirely other possible platforms for testing gravitational decoherence, such as opto-mechanics, or space experiments with photons. How do the authors substantiate their claim? Clearly, a full analysis requires a lot of work, but some rule-of-thumb estimates should not be difficult to give.”

Reply to Comment 4): We are again very grateful to the Reviewer for taking up this truly important point. We agree that the previous discussion of the possible experimental tests was way too poor.

We have now performed a thorough analysis of possible experimental schemes for the verification of our model, and we have introduced an entire new Section, Section V in the current second revision. In this new Section we first review the most significant theoretical measures of decoherence introduced so far in the literature, and we then illustrate an experimental scheme based on an optomechanical setup, with massive ultracold molecules as mechanical oscillators, for the experimental test of our model.

Our findings are that in order to have a relevant and detectable effect the optomechanical apparatus must be kept at very low temperatures of the order of 10^{-16} K with mechanical oscillators having a mass of the order of 10^{-16} Kg. The temperature range falls between three and four orders of magnitude lower than the lowest temperatures currently achievable with mechanical oscillators in cavity optomechanics. It is foreseeable that further progress in cooling techniques will allow to reach the needed temperature scale in the not so far future. Also the required mass range should be achievable in the next few years, according to the most recent extrapolations from current experiments. Concerning the needed mass scale, remarkably it turns out to be of the same order of magnitude required by other experimental schemes aiming at investigating the quantum nature of the gravitational interaction.

In conclusion, we are confident that, with the above further additions and improvements, this second revised version of our manuscript may be now considered suitable for publication in Nature Communications.

Sincerely yours,

Fabrizio Illuminati

Luciano Petruzzello

Reviewers' Comments:

Reviewer #3:

Remarks to the Author:

I thank the authors for the exhaustive reply to my comments. I think we reached an agreement regarding the status of the assumptions defining their model. Although I do not agree with the "naturalness" of the assumptions, future research will tell who is right. I also appreciated the analysis of the potential testability of the model with an optomechanical setting, which I consider a significant improvement of the work. I am happy with the paper as it is now.